# Robust Deep Neural Networks for Heterogeneous Tabular Data

## Abstract

Although deep neural networks (DNNs) constitute the state-of-the-art in many tasks based on image, audio, or text data, their performance on heterogeneous, tabular data is typically inferior to that of decision tree ensembles. To bridge the gap between the difficulty of DNNs to handle tabular data and leverage the flexibility of deep learning under input heterogeneity, we propose *DeepTLF*, a framework for deep tabular learning. The core idea of our method is to transform the heterogeneous input data into homogeneous data to boost the performance of DNNs considerably. For the transformation step, we develop a novel knowledge distillations approach, *TreeDrivenEncoder*, which exploits the structure of decision trees trained on the available heterogeneous data to map the original input vectors onto homogeneous vectors that a DNN can use to improve the predictive performance. Through extensive and challenging experiments on various real-world datasets, we demonstrate that the DeepTLF pipeline leads to higher predictive performance. On average, our framework shows 19.6% performance improvement in comparison to DNNs. The DeepTLF code is publicly available.

## 1 Introduction

Tabular data is the most commonly used form of data, and it is ubiquitous in various applications, such as medical diagnosis based on patient history (Fatima et al., 2017), predictive analytics for financial applications (Dastile et al., 2020), cybersecurity (Buczak & Guven, 2015), and so forth. Although DNNs perform outstandingly well on homogeneous data, e.g., visual, audio, and textual data (Goodfellow et al., 2016), heterogeneous, tabular data still pose a challenge to these models (Shwartz-Ziv & Armon, 2021).

We hypothesize that the moderate performance of DNNs on tabular data comes from two major factors. The first is the inductive bias(es) (Katzir et al., 2020); for example, DNNs assume that certain spatial structures are present in the data (Mitchell et al., 2017), whereas tabular data do not have any spatial connections. The second reason is the high information loss during the data preprocessing step since tabular input data need to undergo cleansing (dealing with missing, noisy, and inconsistent values), uniform discretized representation (handling categorical and continuous values together), and scaling (standardized representation of features) steps. Along with these feature-processing steps, important information contained in the data may get lost, and, hence, the preprocessed feature vectors (especially when one-hot encoded) may negatively impact training and learning effectiveness (García et al., 2015). As reported by Hancock & Khoshgoftaar (2020), an efficient transformation of heterogeneous data for training DNNs is still a significant challenge.

Typically, when heterogeneous tabular data are involved, the first choice across all machine learning (ML) algorithms are ensemble models based on decision trees (Nielsen, 2016), such as random forests (RF) (Breiman, 2001) or gradient-boosted decision trees (GBDT) (Friedman, 2002). Since the inductive bias(es) of the methods based on decision trees are well suited to non-spatial heterogeneous data, the data preprocessing step is reduced to a minimum. In particular, the most common implementations of the GBDT algorithm—XGBoost (Chen & Guestrin, 2016), LightGBM (Ke et al., 2017), and CatBoost (Prokhorenkova et al., 2018) — handle the missing values internally by searching for the best approximation of missing data points.

However, the most significant computational disadvantage of the decision tree–based methods is while training the need to store (almost) the entire dataset in memory (Katzir et al., 2020). Furthermore,

in the multimodal datasets in which different data types are involved (e.g., visual and tabular data), decision tree–based models are not able to provide state-of-the-art results, whereas DNN models allow for batch-learning (no need to store the whole dataset), and for those multimodal data tasks, DNNs demonstrate state-of-the-art performance (Gu & Budhkar, 2021).

Towards the goal of significantly boosting DNNs on tabular data, we propose DeepTLF, a novel deep tabular learning framework that exploits the advantages of the GBDT algorithm as well as the flexibility of DNNs. The key element of the framework is a novel encoding algorithm, TreeDrivenEncoder, which transforms the heterogeneous tabular data into homogeneous data by distilling knowledge from nodes of trained decision trees. Thus, DeepTLF can preserve most of the information that is contained in the original data and encoded in the structure of the decision trees and benefit from preprocessing power of decision tree-based algorithms.

Through experiments on various freely available real-world datasets, we demonstrate the advantages of such a composite learning approach for different prediction tasks. We argue that by transforming heterogeneous tabular data into homogeneous vectors, we can drastically improve the performance of DNNs on tabular data.

The main contributions of this work are: (I) We propose a deep tabular learning framework - DeepTLF that combines the *preprocessing strengths* of GBDTs with the *learning flexibility* of DNNs. (II) The proposed framework builds on a generic approach for transforming heterogeneous tabular data into homogeneous vectors using the structure of decision trees from a gradient boosting model using a novel encoding function – TreeDrivenEncoder. Hence, the transformation approach can also be used independently from the presented deep learning framework. (III) In extensive experiments on various datasets and compared with state-of-the-art ML approaches, we show that the proposed framework mitigates well-known data-processing challenges and leads to unprecedented predictive performance, outperforming all the competitors. (IV) We provide an open-source implementation of DeepTLF https://github.com/xxxx/DeepTLF

## 2 RELATED WORK

In this section, we briefly review the main ideas from prior work that are relevant to our framework and data encoding step. We provide more information on the state-of-the-art DL approaches for tabular data in the Appendix B.

Independent works Moosmann et al. (2007); Geurts et al. (2006) demonstrate that data can be encoded using the RF algorithm by accessing leaf indices in the decision trees. The idea was also utilized by (He et al., 2014), where instead of the RF model, trees from a GBDT model are used for the categorical data encoding. These works demonstrates that the decision trees are a powerful and convenient way to implement non-linear and categorical feature transformations for heterogeneous data. The DeepGBM framework Ke et al. (2019) further evolved the idea of distilling knowledge from decision trees leaf index by encoding them using a neural network for online learning tasks. This approach is quite popular, but *the leaf indices' from a decision tree embedding do not fully represent the whole decision tree structure*. Thus, each boosted tree is treated as a new *meta categorical feature*, which might be an issue for the DNNs (Hancock & Khoshgoftaar, 2020).

In contrast to related methods, our aim is to *holistically* distill the information from decision trees, by utilizes the whole decision tree, not only the output leafs. The DeepTLF combines the advantages of GBDT (such as handling missing values, categorical variables) with the learning flexibility of DNNs to achieve superior and robust prediction performance. Also, Medvedev & D'yakonov (2020) demonstrated that a DNN which was trained using distilled data can outperform models trained on the whole original data.

Other approaches such as NODE (Popov et al., 2019), Net-DNF (Katzir et al., 2020) try to mimic the decision trees using DNNs. A difference to our framework is that DeepTLF is more robust to the data inconsistencies and does not require new DNN architectures, hence, it is straightforward to use.

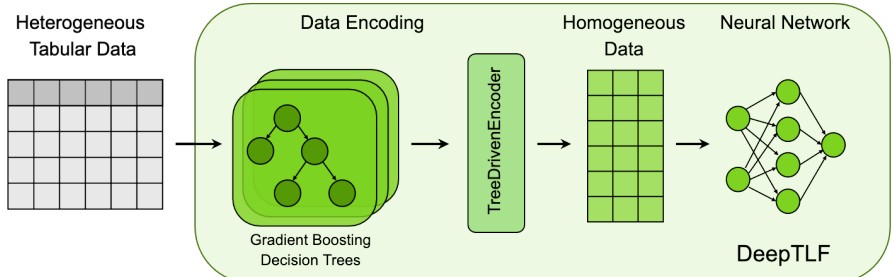

Figure 1: A data pipeline for the DeepTLF framework. First, the original tabular data is used to train a gradient boosted decision trees (GBDT) model. The heterogeneous data (i.e., training as well as the test data) are transformed by exploiting the structures of the decision trees in the ensemble. More specifically, the *TreeDrivenEncoder* algorithm distills information from trained decision trees of the GBDT model to produce homogeneous binary vectors. These vectors are then used to train a DNN. Note that DeepTLF does not require data preprocessing (such as normalization or handling missing or categorical values); therefore, in total, it dramatically speeds up the *data preprocessing* time.

## 3  DEEPTLF: DEEP TABULAR LEARNING FRAMEWORK

In this section, we present the main components of our DeepTLF framework. As shown in Fig. 1, DeepTLF consists of three major components: (1) an ensemble of decision trees (in this work, we utilize the GBDT algorithm), (2) a TreeDrivenEncoder that performs the transformation of the original data into homogeneous, binary feature vectors by distilling the information contained in the structures of the decision trees through the TreeDrivenEncoder algorithm, and (3) a deep neural network model trained on the binary feature vectors obtained from the TreeDrivenEncoder algorithm. We will describe the details of each component in the following subsections.

### 3.1  GRADIENT BOOSTED DECISION TREE

For the data encoding step, we selected one of the most powerful algorithms on tabular data, namely the gradient boosted Decision Trees (GBDT) algorithm (Friedman, 2002). GBDT is a well-known and widely used ensemble algorithm for tabular data both in research and industrial applications (Chen & Guestrin, 2016) and is particularly successful for tasks containing heterogeneous features, small dataset sizes, and "noisy" data (Nielsen, 2016). Especially when it comes to handling variance and bias, gradient boosting ensembles show highly competitive performance in comparison with state-of-the-art learning approaches (Friedman, 2002; Nielsen, 2016). In addition, multiple evaluations have empirically demonstrated that the *decision trees of a GBDT ensemble preserve the information* from the original data and can be used for further data processing (He et al., 2014; Ke et al., 2019).

The key idea of the GBDT algorithm is to construct a strong model by iterative addition of weak learners. Formally, at each iteration $k$ of the gradient boosting algorithm, the GBDT model $\varphi$ can be defined as:

$$\varphi^k(\mathbf{x}) = \varphi^{k-1}(\mathbf{x}) + \lambda\, h^k(\mathbf{x}), \tag{1}$$

where $\mathbf{x}$ is an input feature vector, $\varphi^{k-1}$ is the strong model constructed at the previous iteration, $h$ is a weak learner from a family of functions $\mathcal{H}$, and $\lambda$ is the learning rate.

$$h^k = \arg\min_{h \in \mathcal{H}} \sum_i \left( -\frac{\partial L(\varphi^{k-1}(\mathbf{x}_i), y_i)}{\partial \varphi^{k-1}(\mathbf{x}_i)} - h(\mathbf{x}_i) \right)^2. \tag{2}$$

More specifically, a pseudo-residual $-\frac{\partial L(\varphi^{k-1}(\mathbf{x}_i), y_i)}{\partial \varphi^{k-1}(\mathbf{x}_i)}$ should be approximated as well as possible by the current weak model $h(\mathbf{x}_i)$. The gradient w.r.t. the current predictions indicate how these predictions should be changed in order to minimize the loss function. Informally, gradient boosting can be thought of as performing gradient descent in the functional space.

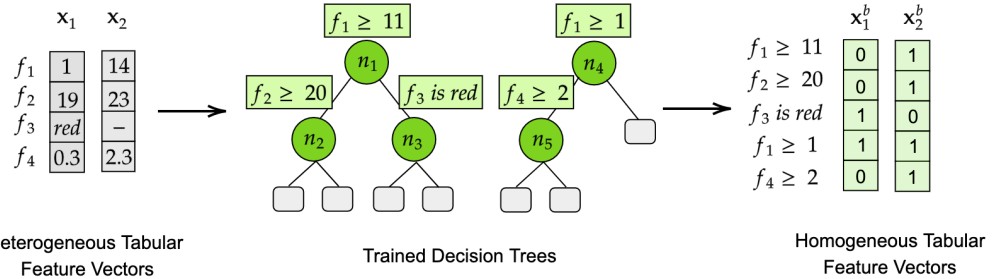

Figure 2: A toy example of the proposed data transformation performed by the *TreeDrivenEncoder* algorithm. On the left, we see two heterogeneous input feature vectors $\mathbf{x}_1$ and $\mathbf{x}_2$ from the original dataset $\mathcal{D}$, where $\mathbf{x}_i \in \mathbb{R}^4$, $f_1, f_2, f_4$ are numerical features, and $f_3$ is a categorical feature in a tabular dataset. Note there is also a missing value $-$ for the feature $f_3$ in $\mathbf{x}_2$. To encode the input data, we use two trained decision trees on the dataset $\mathcal{D}$, with 5 inner nodes in total. By evaluating the Boolean function in each inner node for a given input vector, we construct two homogeneous feature vectors $\mathbf{x}_1^b$ and $\mathbf{x}_2^b$, where a component of these vectors is set to 1 if the corresponding Boolean function evaluates to true and 0 otherwise.

The set of weak learners $\mathcal{H}$ is usually formed by shallow decision trees, which are directly trained on the original data. Consequently, almost no data preparation is needed, and the information loss is minimized. We denote a GBDT model as a set of decision trees:

$$GB(\mathbf{X}, \mathbf{y}) = \{T_1, T_2, ..., T_k\},$$

where $\mathbf{X}$ is a matrix of observations, $\mathbf{y}$ is a vector or matrix of labels.

### 3.2 KNOWLEDGE DISTILLATION FROM GRADIENT BOOSTED DECISION TREES USING TREEDRIVENENCODER

The trained GBDT model provides structural data information, which also encodes dependencies between the input features with respect to the prediction task. In order to distill the knowledge from a tree-based model, we propose a novel data transformation algorithm – *TreeDrivenEncoder*. For every input vector from the original data, the proposed encoding method maps all features occurring in the decision trees of the GBDT ensemble to a binary feature vector $\mathbf{x}^b$. This has the advantage that the neural network in the final component can form its own feature representations from homogeneous data. In Fig. 2, we illustrate the transformation obtained by applying the TreeDrivenEncoder algorithm on a toy example. There we have two input feature vectors $\mathbf{x}_1$ and $\mathbf{x}_2$ with categorical and numerical values are encoded into corresponding homogeneous binary feature vectors $\mathbf{x}_1^b$ and $\mathbf{x}_2^b$.

To formally describe the TreeDrivenEncoder algorithm, we first need a definition of the decision trees:

**Definition 1 (Decision Tree)** *Let $T$ be a structure $T = (V, E, \mu)$, where $V$ is a set of nodes, $E \subseteq V \times V$ is a set of edges and $\mu = (\mu_v)_{v \in V}$ is a sequence of mapping functions $\mu_v : \mathbb{R}^d \to V \cup \{\emptyset\}$ that map input vectors to (child) nodes. We call $T$ a (binary) decision tree if it satisfies the following properties:*

1. *$(V, E)$ is a directed acyclic graph*

2. *There is exactly one designated node $v_r \in V$, called the root, which has no entering edges, i.e. for a node $v \in V$:*
$$v = v_r \Leftrightarrow \forall w \in V : (w, v) \notin E.$$

3. *Every node $v \in V \backslash \{v_r\}$ has exactly one entering edge with the parent node at its other end:*
$$w \in V : (w, v) \in E \Leftrightarrow w = parent(v).$$

4. *Each node has either two or zero outgoing edges. We call the nodes with two outgoing edges **inner nodes** and all others nodes **leaves**. We denote the sets of inner nodes and leaves with $V_I$ and $V_L$, respectively.*

5. $\mu_v$ *maps feature vectors from inner nodes to their child nodes and from leaves to* $\emptyset$.

$$v \in V_I \Rightarrow \forall \mathbf{x} \in \mathbb{R}^d : (v, \mu_v(\mathbf{x})) \in E, \tag{3}$$

$$v \in V_L \Rightarrow \forall \mathbf{x} \in \mathbb{R}^d : \mu_v(\mathbf{x}) = \emptyset. \tag{4}$$

*In the following, we denote the number of inner nodes as* $|T| = |V_I|$. *Furthermore, we assume that the child nodes can be identified as left or right child. For each inner node* $v \in V_I$, *we use a modified mapping function* $\tilde{\mu}_v : \mathbb{R}^d \to \{0, 1\}$ *(i.e., a Boolean function) where* 0 *encodes the left child and* 1 *encodes the right child.*

For an input vector $\mathbf{x} \in \mathbb{R}^d$, we exploit the structure of $T$ to derive a binary vector of length $|T|$. To this end, as shown in Alg. 1, we employ a breadth-first-search approach on the nodes of $T$. More specifically, for every feature that is evaluated at an inner node $v$ of $T$, we retrieve the corresponding value from $\mathbf{x}$ and evaluate that value at $v$ based on the associated Boolean function. Note that other node visiting strategies (e.g., depth-first search) can be used as well. It is only important that the same strategy is used across all decision trees and vectors.

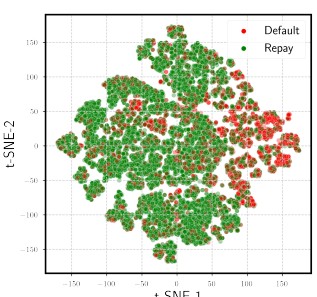

Finally, we concatenate all the vectors generated from the single decision trees of the ensemble $\mathcal{T}$ on the input vector $\mathbf{x}$, which gives us the final binary representation $\mathbf{x}^b$ of $\mathbf{x}$. We kindly ask the Reader to refer to the supplementary materials - we summarize the full algorithm in Alg. 1,

For mathematical completeness, the mapping obtained by applying TreeDrivenEncoder is formalized as follows. Given the feature vector $\mathbf{x}$ that represents an instance from the training dataset $\mathcal{D}$ and a trained decision tree ensemble $\mathcal{T}$ (i.e., a collection of decision trees) on the same dataset, we exploit the structure of each tree $T \in \mathcal{T}$ to produce a binary feature vector for the original feature vector $\mathbf{x} = (x_1, ..., x_d)$ and employ a transformation function:

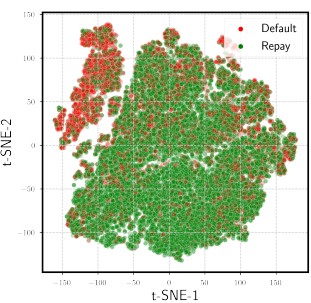

$$map_T : \mathbb{R}^d \to \{0, 1\}^{|T|}, \tag{5}$$

$$map_T : \mathbf{x} \mapsto (\tilde{\mu}_v(\mathbf{x}))_{v \in V_I}, \tag{6}$$

where $V_I$ again represents the inner nodes in a well-defined order and $|T|$ their number. The mapping is performed such that at an inner node $v$ of $T$, the corresponding component $x_j$ of $\mathbf{x}$ *is mapped to 1 if the Boolean function at* $v$ *evaluates to true for* $x_j$ *and 0 otherwise.* Note that we apply the transformation function to each node in the decision tree $T$, even if a node does not belong to the *decision path* of $\mathbf{x}$, hence, it holds that $map_T(\mathbf{x}) \in \{0, 1\}^{|T|}$.

Figure 3: t-SNE visualizations of original heterogeneous tabular default of clients dataset (top), and the same dataset after the TreeDrivenEncoder transformation (bottom).

For the multiple decision trees $T_1, ..., T_k$ we construct a function

$$\texttt{TreeDrivenEncoder} : \mathbb{R}^d \to \{0, 1\}^{\sum_{i=1}^{k} |T_i|} \tag{7}$$

with

$$\texttt{TreeDrivenEncoder}(\mathbf{x}) = (map_{T_1}(\mathbf{x})...map_{T_k}(\mathbf{x})) \tag{8}$$

### 3.3 Deep Learning Models for Encoded Homogeneous Data

After the data distillation by the TreeDrivenEncoder algorithm, the new binary representations of the feature vectors are used to train and validate a chosen neural network model.

## 4 Experiments

To evaluate the performance of DeepTLF against state-of-the-art models, we employ several real-world heterogeneous datasets of varying sizes from different application domains. In the following,

Table 1: Details of the datasets used in the experimental evaluations. #Sample is the number of data points, #Num is the number of numerical variables, and #Cat is the number of categorical variables in a dataset.

|    | Dataset | #Samples | #Num | #Cat | Task |
|----|---------|----------|------|------|------|
| D1 | HIGGS | 11,000,000 | 28 | 0 | Binary classification |
| D2 | Default of clients | 30,000 | 14 | 9 | Binary classification |
| D3 | Telecom churn | 51,047 | 38 | 18 | Binary classification |
| D4 | Zillow | 167,888 | 31 | 27 | Regression |
| D5 | Avocado prices | 18,249 | 8 | 3 | Regression |
| D6 | California housing | 20,640 | 8 | 0 | Regression |
| D7 | E-commerce clothing reviews | 23,486 | 6 | 4 | Binary classification |

Table 2: Experimental results based on (stratified) 5-fold cross-validation. We use the same fold splitting strategy for every dataset. The cross-entropy measure (lower is better) is selected for classification tasks and MSE measure (lower is better) is selected for regression problems respectively. The top results for each dataset are marked in bold.

| | Classification Datasets | | | Regression Datasets | | |
|---|---|---|---|---|---|---|
| | D1 | D2 | D3 | D4 | D5 | D6 |
| LR | 0.637±0.001 | 0.470±0.001 | 0.584±0.001 | 0.028±0.001 | 0.966±0.001 | 0.552±0.063 |
| RF | 0.502±0.001 | 0.444±0.007 | 0.564±0.003 | 0.028±0.001 | 0.025±0.001 | 0.254±0.008 |
| GBDT | 0.498±0.001 | 0.429±0.006 | 0.559±0.003 | 0.027±0.003 | 0.026±0.003 | 0.217±0.021 |
| Leafs+LR | 0.659±0.001 | 0.453±0.002 | 0.580±0.002 | 0.029±0.001 | 0.105±0.036 | 0.358±0.032 |
| | Deep Neural Network Models | | | | | |
| DNN | 0.511±0.001 | 0.437±0.005 | 0.579±0.002 | 0.028±0.001 | 0.069±0.002 | 0.339±0.122 |
| RLN | 0.507±0.002 | 0.433±0.051 | 0.599±0.001 | 0.399±0.042 | 0.275±0.244 | 0.947±0.228 |
| DeepGBM | 0.487±0.001 | 0.457±0.023 | 0.589±0.001 | **0.026±0.001** | 0.038±0.045 | 0.299±0.017 |
| TabNet | 0.503±0.001 | 0.447±0.001 | 0.591±0.005 | 0.049±0.001 | 0.073±0.002 | 0.455±0.106 |
| VIME | 0.514±0.001 | 0.453±0.006 | 0.593±0.002 | 0.030±0.003 | 0.120±0.016 | 0.684±0.023 |
| TabTransformer | 0.581±0.002 | 0.515±0.003 | 0.650±0.021 | 0.029±0.001 | 0.073±0.002 | 0.994±0.501 |
| TabNet | 0.503±0.001 | 0.447±0.001 | 0.591±0.005 | 0.049±0.001 | 0.073±0.002 | 0.455±0.106 |
| Net-DNF | 0.561±0.001 | 0.512±0.001 | 0.594±0.003 | 0.027±0.001 | 0.321±0.093 | 2.491±0.051 |
| NODE | 0.489±0.006 | 0.458±0.006 | 0.598±0.001 | 0.028±0.001 | 0.104±0.030 | 0.722±0.052 |
| **DeepTLF (ours)** | **0.483±0.001** | **0.427±0.006** | **0.557±0.003** | **0.026±0.001** | **0.021±0.005** | **0.215±0.012** |

we first provide details about our experimental setup, including a description of the datasets. We then compare the performance between different ML models and DeepTLF. After, we present a training, inference, and data preprocessing runtime analysis. For the purpose of reproducibility we provide technical details of the experiments. Furthermore, in the Appendix A, we provide more empirical results to support our work.

## 4.1 EXPERIMENTAL SETTINGS

### 4.1.1 DATASETS

For the evaluation of DeepTLF, we used six heterogeneous and one multimodal dataset from different domains as described in Table 4, each dataset was previously featured in multiple published studies. The web access points and disruptions of each dataset are in the Appendix D.1. The data is pre-processed in the same way for each experiment; we do normalization and missing values subsection steps, except for GBDT and DeepTLF; since these approaches can handle missing values independently.

### 4.1.2 BASELINE MODELS

For the baseline models, we select the following algorithms: LR, linear or logistic regression models; Random Forest (RF) (Breiman, 2001); for GBDT (Friedman, 2002), we utilize the XGBoost implementation (Chen & Guestrin, 2016); DNN, A deep neural network with three fully-connected layers and two DropOut layers (Srivastava et al., 2014); Leafs+LR, A hybrid model, combining leaf

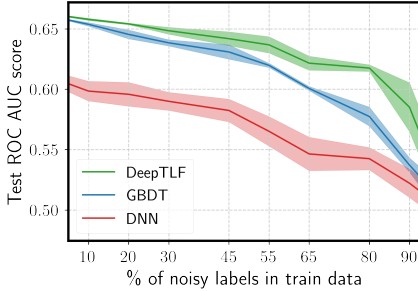 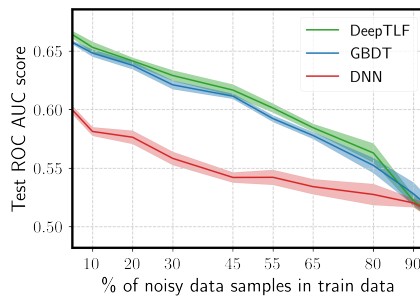

Figure 4: Left: Noisy labels experiment. Right: Noisy data experiment. The DNN here is identical to the DL part in the DeepTLF. Note the test data is not corrupted. We report the ROC AUC value (higher is better). Results are averages over five trials for the telecom churn (D3) dataset.

index from a trained GBDT model and generalized linear models proposed by He et al. (2014); RLNs (Shavitt & Segal, 2018), Regularization Learning Networks (RLNs) is a dedicated to tabular learning DNN, which uses the counterfactual loss to tune its regularization hyperparameters efficiently. TabNet (Arik & Pfister, 2019) is a deep tabular data learning architecture, which uses sequential attention to choose which features to reason from at each decision step; Neural Oblivious Decision Ensembles (NODE) (Popov et al., 2019) is a deep tabular data learning architecture, which generalizes ensembles of oblivious decision trees, but benefits from both end-to-end gradient-based optimization and the power of multilayer hierarchical representation learning; DeepGBM (Ke et al., 2019), a deep learning framework distilled by the GBDT algorithm; Net-DNF (Katzir et al., 2020); VIME (Yoon et al., 2020), a self-supervised learning framework for tabular data; TabTransformer (Huang et al., 2020), a framework built using self-attention transformers; Lastly, DeepTLF (*the proposed algorithm*), consisting of a four fully-connected layers with the two DropOut layers to lower the overfitting effect. We *deliberately* select a relatively simple neural network model without advanced layers such as batch normalization or attention (transformer) to demonstrate the power of our approach. By applying more sophisticated DL techniques, the model performance can be further improved.

## 4.2 PERFORMANCE EVALUATION

**Main benchmark.** In our performance evaluation, we partitioned each of the datasets using *(stratified) 5-fold cross-validation*. We use the following quality measures cross-entropy loss for classification tasks and mean-squared error (MSE) for regression tasks. Results are reported in terms of mean and standard deviation values in Table 2.

**Corrupted data.** We also compare the performance of DeepTLF with a plain DNN and GBDT under corrupted data to verify how the proposed model performs in scenarios of noisy labels and noisy data in the training dataset (Fig. 4).

*Noisy training data and labels.* We use two different setups: noisy training labels and noisy training data. We artificially corrupted the customer churn dataset (Sec. 4.1.1) by introducing random noise either to the training labels (labels were shuffled) and the training dataset. Note that for validation purposes, the test dataset was not corrupted. A distinguishing strength of the DeepTLF framework compared to other state-of-the-art approaches in the field is that it can handle missing values internally through the proposed gradient-boosting embeddings.

*Missing values experiment.* Figures 9 in the Appendix show the performance of DNN, GBDT, and DeepTLF models with different proportions of missing in the training dataset. As we can see, the performance of the DNNs drops drastically, while DeepTLF shows stable performance.

**Sensitivity to hyperparameters.** This experiment demonstrates how the GBDT hyperparameters contribute to the final performance of the DeepTLF such as the number of decision trees (Fig. 5) and learning rate (Fig. 10). For comparison purposes, we also add the GBDT baseline to the figures. It can be seen that DeepTLF does not require extensive hyperparameter tuning, since it reaches the saturation level.

**t-SNE visualizations.** We also compare t-SNE visualizations (Van der Maaten & Hinton, 2008) of the default of the clients dataset and a TreeDrivenEncoder encoded version of the same dataset,

Table 3: A comparison of the training and inference runtime for selected models from different categories on the whole Zillow dataset (167,888 samples). The results related to the training, inference and preprocessing time are averages over five runs over the whole dataset for training and inference tests. The data preprocessing step includes: data scalling, handling missing values.

| Model | Training time (s) | Inference time (s) | Data preprocessing time (s) | #Learning parameters |
|---|---|---|---|---|
| GBDT (CPU) | 13.5 | 0.5 | **0** | 200 trees, depth 4 |
| GBDT (GPU) | **3.1** | **0.3** | **0** | 200 trees, depth 4 |
| DeepGBM (GPU) | 23.9 | 5.2 | 0.3 | 222,548 weights |
| TabNet (GPU) | 79.1 | **2.2** | 0.3 | 584,832 weights |
| NODE (GPU) | 310.2 | 15.5 | 0.3 | 27,105,922 weights |
| DeepTLF (GPU) | **15.1** | 3.2 | **0** | 80,351 weights |

the results are shown in Fig. 3. It can be seen that TreeDrivenEncoder indeed preserves valuable information from the trained decision trees.

**Multimodal data.** In this experiment, we demonstrate how our framework performs on multimodal data. We select the e-commerce clothing reviews dataset (Agarap, 2018), which *has two data modalities textual and tabular data*, and compare to DNN and DeepTLF models on unseen validation data (Fig. 6). The only difference between the DNN and DeepTLF models in this experiment is the tabular data representation, for DNN it is the original heterogeneous dataset, where the proposed framework utilizes TreeDrivenEncoder for the data transformation step. The results demonstrate the efficiency of our framework in the multimodal setting.

**Training/Inference Runtime Comparison.** Finally, we compare the runtime performance between several DL-based algorithms with GBDT (XGBoost (Chen & Guestrin, 2016)). Table 3 summarizes our results. To make a fair comparison, we used the latest available versions of the corresponding implementations. Also, we utilize the same DL framework, PyTorch (Paszke et al., 2017), and the same number of epochs for each DL-based baseline. One of the possible reasons for the gap between the proposed method and other DL-based approaches is that DeepTLF utilizes a simple deep neural network, whereas other approaches apply transformer networks or specialized decision tree-like layers. We also report the data preprocessing time for each baseline.

## 5 DISCUSSION

**Empirical evaluations.** We can derive the following observations from all experiments: Our framework, DeepTFL, combines the preprocessing strengths of gradient-enhanced decision trees with the learning flexibility of deep neural networks. It can handle heterogeneity in the data very well and hence shows to be highly efficient. Also, it shows a stable performance independently of data size. Moreover, the proposed encoding algorithm can be employed for *multimodal learning problems* Ngiam et al. (2011); Gu & Budhkar (2021), where multimodal data involve both tabular and other data sources (text, image, sound) in an integrated manner (i.e., end-to-end deep learning) while achieving a robust performance. Finally, with regard to data quality issues (noisy data and labels, misssing values), our approach clearly outperforms the DNN and GBDT models, thus showing *to be applicable to many real-world applications where data loss occurs frequently.*

**Decision tree model choice.** Noteworthy, the proposed prediction approach can use any decision tree ensemble as a basic algorithm; in this work, we adopted GBDT

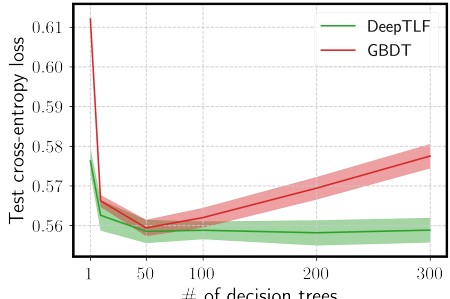

Figure 5: A relationship between a number of trees in the GBDT and the final performance of the proposed DeepTLF framework. The precise same GBDT model is used for the data encoding in the DeepTLF. Results are averaged over five trials for the D3 dataset.

because of its well-known superior performance and its robust feature handling capacities. In addition,

the GBDT algorithm sequentially constructs the trees; at each step, the next tree maximally reduces the loss given the current loss. Thus, *there are conditional dependencies between the trees* in the GBDT ensemble, and as a consequence, they provide *adequate coverage of the data distribution*.

**Hyperparameter selection for DeepTLF.** In our experiments, we demonstrate that the DeepTLF framework does not required extensive tinning for the of the decision tree ensemble part (Fig. 5 and Fig. 8), after reaching the saturation level, the number of trees does not have significant effect.

**Tabular data encoding.** Besides constructing a new homogeneous representation for the heterogeneous, tabular data, *TreeDrivenEncoder* encodes information about the whole dataset, as represented by the structures of the decision trees, which can be seen as a *local feature selection* (and feature engineering). Furthermore, in terms of efficient representation, the encoded binary data has a drastically smaller size than the original heterogeneous data, since real-valued features are typically represented as 32-bit float types. In contrast, a binary vector can be efficiently represented by a sequence of Boolean values (i.e., 1 bit per value). This allows for efficient training in the final component for the DeepTLF model.

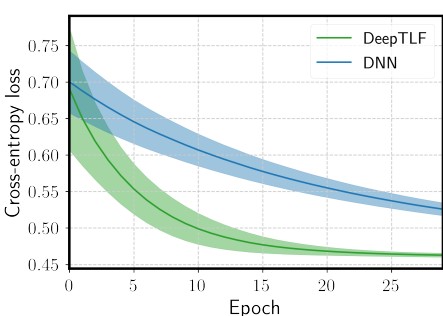

Figure 6: A *multimodal* data experiment on the textual and tabular from the D7 dataset, the DNN and DeepTLF have identical DL architectures and training setups. Results are averaged over five trials.

**Future work and limitations**. Further analysis is needed to investigate the performance of DeepTLF in online learning scenarios. We also see further potential in improving the efficiency of DeepTLF by replacing the decision trees with an efficient neural net transformation layer, thus achieving an end-to-end deep learning mechanism for heterogeneous and multimodal data. Further improvements of our approach could be the usage of more advanced deep learning architectures such as convolution or attention-based (Transformers) neural networks.

## 6 CONCLUSION

In this work, we discussed the challenge of learning from heterogeneous tabular data with deep neural networks. The challenge stems from the concurrent existence of numerical and categorical feature types, complex, irregular dependencies between the features, and other data-related issues such as scales, outliers, and missing values. To address the challenge, we proposed DeepTLF, a framework that exploits the decision trees' structures from an ensemble model to map the original data into a homogeneous feature space where deep neural networks can be effectively and robustly trained. This allows DeepTLF to distill and conserve relevant information in the original data and utilize it in the deep-learning process. Furthermore, the distillation step reduces the required preprocessing to a minimum and can mitigate the mentioned data-related issues by exploiting decision trees' data-processing advantages (internal handling, missing values, and data scaling). Our extensive empirical evaluation on real-world datasets of different sizes and modalities convincingly showed that DeepTLF consistently outperforms the evaluated competitors, which are state-of-the-art approaches in this field. Also, the proposed framework showed robust performance on corrupted data (noisy labels, noisy data, and missing values). Compared to most approaches in this field, DeepTLF is easy to use and does not require changes to existing ML pipelines, which is essential for many practical applications. Moreover, we provide an open-source implementation of DeepTLF which can be used researchers and practitioners for various learning tasks on heterogeneous or multimodal tabular data.

## 7 ETHICS STATEMENT

The proposed deep tabular framework and data encoding algorithm closes the performance gap between deep learning and tree-based ensemble approaches, such as gradient boosting decision trees, on heterogeneous tabular data. Thus, with DeepTLF, well-studied deep learning and decision-tree-based approaches, along with privacy-preserving mechanisms, can be also applied to challenging

settings related to heterogeneous tabular data. Moreover, the homogeneous decision tree encoded vectors from the DeepTLF algorithm can be used for further interpretation tasks by implementing importance analysis using SHAP (Lundberg & Lee, 2017) or LIME (Ribeiro et al., 2016) algorithms.

## 8 REPRODUCIBILITY DETAILS

For the purpose of reproducibility, we provide details of experiment settings, including hyperparameter ranges for tuning, and access points for datasets of the study are available in Appendix D. Besides, we also make publicly available our implementation of the DeepTLF framework with the scoring function which we used in this study.

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

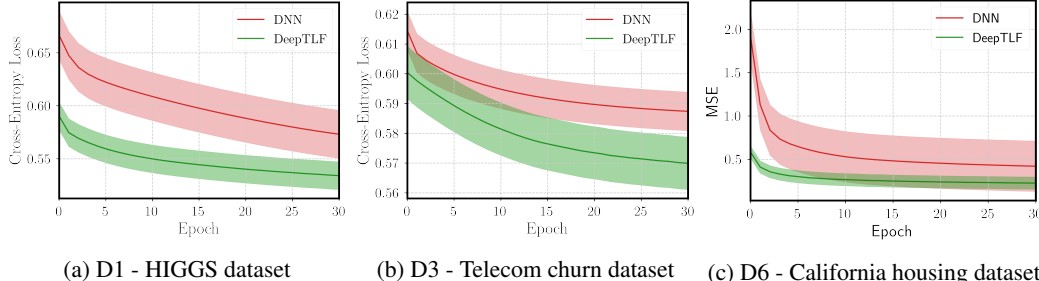

(a) D1 - HIGGS dataset      (b) D3 - Telecom churn dataset      (c) D6 - California housing dataset

Figure 7: The comparison of the DeepTLF (a green line) and the deep neural model (DNN) (a red line) models on validation (unseen) data. DeepTLF and DNN models have the exact same architecture. The results are computed over ten runs with different random seeds.

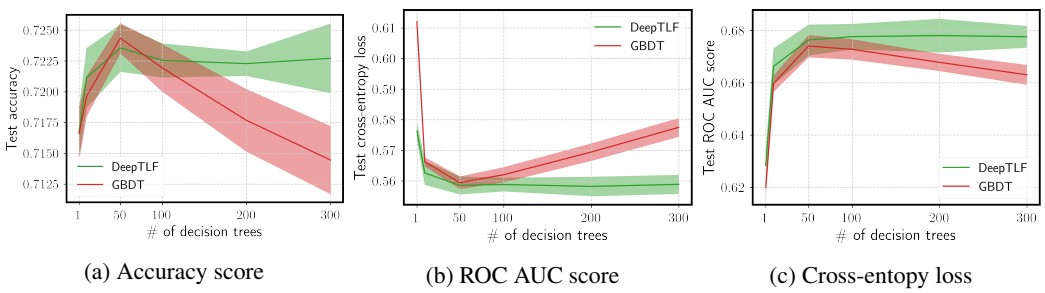

(a) Accuracy score      (b) ROC AUC score      (c) Cross-entopy loss

Figure 8: A relationship between number of decision trees and the DeepTLF performance. The accuracy score (higher is better), ROC AUC score (higher is better), cross-entropy loss (lower is better) metrics for the same experiment. The exact same GBDT model is used for the data encoding in the DeepTLF. The results are averages over five trials for the telecom churn (D3) dataset.

## A    ADDITIONAL EXPERIMENTS

In this Section, we provide more experimental results.

**Validation loss curves.** We examine DNNs and our DeepTLF model separately using only the validation (unseen) data. To enable a fair comparison, the deep learning part in DeepTLF is identical to the DNN we used. The results are presented in Fig. 7.

**Is there a correlation between GBDT's performance and the final performance of the DeepTLF?** In this experiment, we examine the correlation between the performance of GBDT (which is used for data encoding) and DeepTLF. In other words, we want to demonstrate that if the performance of the GBDT improves, the performance of the DeepTLF rises. Fig. 11 presents the results of the experiments; as it can be observed, there is indeed a high positive correlation between the performance of the GBDT and DeepTLF. It is also noticeable that the DeepTLF performance in many cased the GBDT results.

**A "sanity check" experiment.** In this simple experiment, we want to verify that our heterogeneous encoding function distills knowledge better than a random heterogeneous "encoding function". We design the random function in the what it extracts random features $f$ with random splitting value. The experiment confirms that indeed the proposed encoder performs better than a random set of Boolean rules for a given dataset (Fig. 12).

## B    EXTENDED RELATED WORK

In recent years, deep learning on tabular data has received much attention from the machine learning and data science communities (Arik & Pfister, 2019; Popov et al., 2019; Yin et al., 2020; Huang et al., 2020; Guo et al., 2017; Ke et al., 2019; He et al., 2014; Shavitt & Segal, 2018; Katzir et al., 2020; Yoon et al., 2020; Padhi et al., 2020; Levy et al., 2020; Ballet et al., 2019; Akrami et al., 2020; Gupta

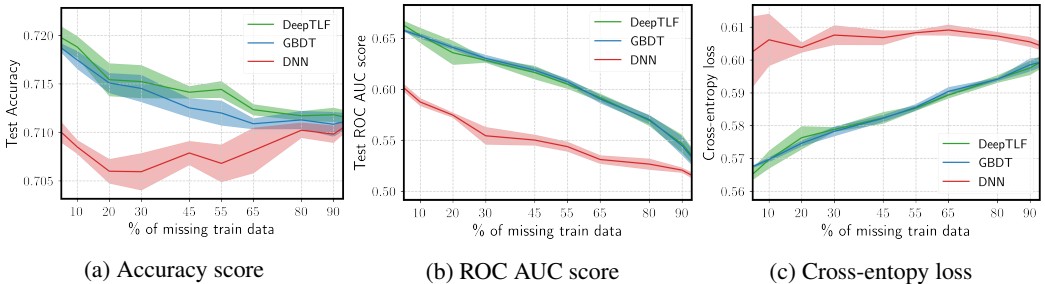

(a) Accuracy score       (b) ROC AUC score       (c) Cross-entropy loss

Figure 9: The missing data experiment. The accuracy score (higher is better), ROC AUC score (higher is better), cross-entopy loss (lower is better) metrics for the same experiment. The exact same GBDT model is used for the data encoding in the DeepTLF. The DNN model is identical in training and architecture to the DeepTLF's DNN part. The results are averages over five trials for the telecom churn (D3) dataset.

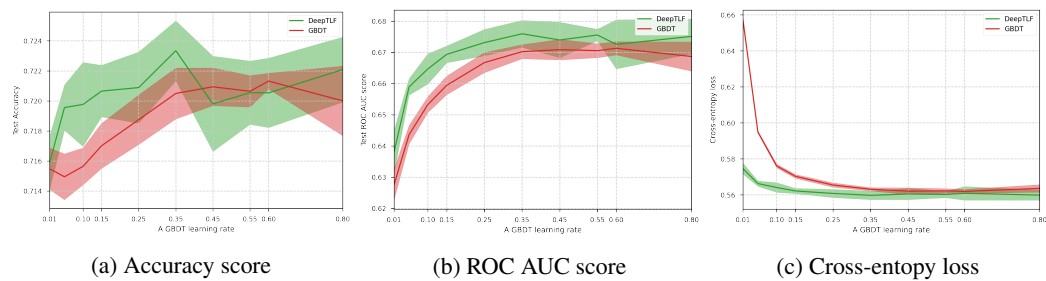

(a) Accuracy score       (b) ROC AUC score       (c) Cross-entropy loss

Figure 10: Learning rate. The results are averages over five trials for the telecom churn (D3) dataset.

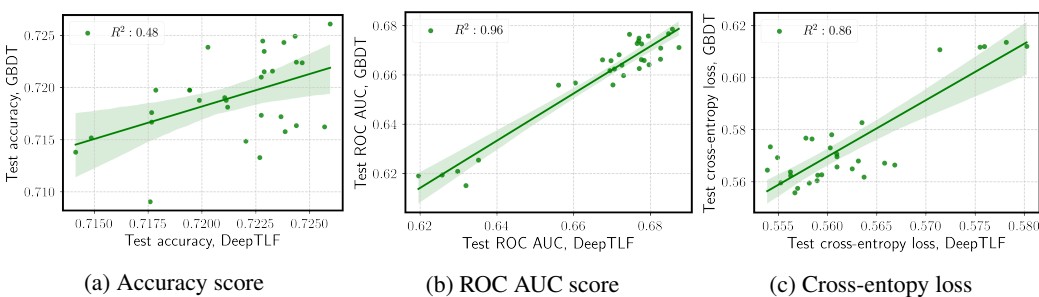

(a) Accuracy score       (b) ROC AUC score       (c) Cross-entropy loss

Figure 11: Correlation plots for different quality measurements. The exact same GBDT model is used for the data encoding in the DeepTLF. The results demonstrate that there is indeed a high positive relationship between the performance of GBDT and DeepTLF. Thus, the proposed data distillation algorithm can successfully distill the knowledge from trees. The results are averaged over five trials for the telecom churn (D3) dataset.

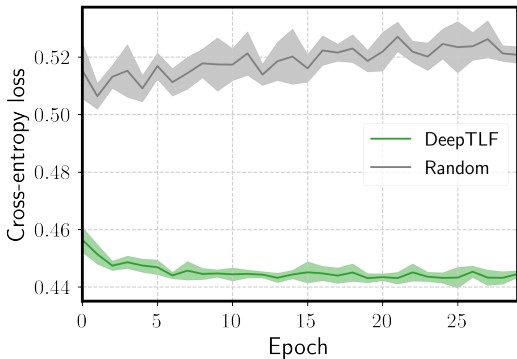

Figure 12: A "sanity check" experiment. A comparison of the TreeDrivenEncoder and random encoding functions. The random encoding mimics the TreeDrivenEncoder, but it selects a random feature and splitting value. The experiment verifies that the TreeDrivenEncoder is able to distill the knowledge using trained decision trees in a GBDT algorithm. The results are averaged over five trials.

et al., 2021). The existing approaches can be grouped into two broad categories - architecture-based and data transformation-based models.

**Architecture-based models.** This group aims at developing new deep learning architectures for heterogeneous data (Arik & Pfister, 2019; Popov et al., 2019; Guo et al., 2017; Ke et al., 2019; Shavitt & Segal, 2018). For example, the authors of (Guo et al., 2017) proposed distinct neural network architecture for reducing the preprocessing and feature engineering effort by introducing a data sharing strategy between a deep and a wide network so that low- and high-level interactions between the inputs can be learned simultaneously, based on the ideas of Factorization Machines (FM) proposed in (Rendle, 2010). The work (Lian et al., 2018) extended the sharing strategy using the FM for structured data further. In (Ke et al., 2019), the authors propose an integrated solution by introducing two special neural networks, one for handling categorical features and another for numerical data. However, for mentioned approaches ((Guo et al., 2017; Lian et al., 2018; Ke et al., 2019)), it is not clear how other data-related issues, such as missing values, different scaling of numeric features, and noise, influence the predictions produced by the models.

Another line of research in this group tries to combine the advantages of decision trees and neural networks. For example, the authors of (Rota Bulo & Kontschieder, 2014) introduced Neural Decision Forests, an ensemble of neural decision trees, where split functions in each tree node are randomized Multi-Layer Perceptrons (MLPs). Another approach (Denoyer & Gallinari, 2014) presented a strategy for selecting paths in a neural directed acyclic graph to produce the prediction for a given input. Hence, the selected neural paths are specialized to specific inputs. In (Wang et al., 2017), the authors empirically showed that neural networks with random forest structure could have better generalization ability across various input domains.

A fully differentiable architecture for deep learning, which generalizes ensembles of oblivious decision trees on tabular, is also introduced in (Popov et al., 2019). Their architecture (coined NODE) employs the entmax transformation (Peters et al., 2019) and thus maps a vector of real-valued scores to a discrete probability distribution.

Other approaches focus on architectures that build on attention-based (Transformers) mechanisms (Vaswani et al., 2017). For example, the authors of (Arik & Pfister, 2019) and (Huang et al., 2020) propose an attentive transformer architecture for deep tabular learning. Their architecture also offers the possibility to interpret the input features; however, for reliable performance, a large amount of training data is needed. Another drawback is that the attention mechanism is only applied to categorical data. Hence, the continuous data does not throw the self-attention block, meaning that correlations between categorical and continuous features are dropped. The work (Yoon et al., 2020) proposes a variation of a transformer and offers semi-supervised learning.

However, no clear statements can be drawn for all methods described so far regarding the relationship between data heterogeneity and prediction quality (especially robustness under noisy data or labels).

Moreover, many of the solutions in this line of research are quite challenging from a practical perspective since it is often unclear which architectural choices should be employed in realistic scenarios. These architecture-based approaches generally rely on novel neural network architectures, which are difficult to (re-)implement and optimize for specific real-world use cases. Also, the work (**?**) promotes localized decisions that are taken over small subsets of the features. Especially for critical, data-intensive applications, e.g., data streaming, large-scale recommendation systems (Baylor et al., 2017), and many more, it is not clear what additional adjustments to the working pipeline are needed.

**Data transformation-based models.** Another way for improving the predictive quality in the presence of tabular data is to transform heterogeneous data into homogeneous feature vectors. The transformation can range from simple data preprocessing, such as the normalization of numerical variables or binary encoding of categorical variables, to linear or non-linear embedding schemes (e.g., generated by advanced autoencoders) (García et al., 2015; Hancock & Khoshgoftaar, 2020). The advantage of such data transformation approaches is that they do not require adapting the deep learning architecture. However, they may reduce the information content by smoothing critical values that might have been highly relevant for the final prediction.

The method proposed by Moosmann et al. (2007) showed how the data could be encoded using the Random Forests (RF) algorithm by accessing leaf indices in the decision trees. The same idea was presented in (He et al., 2014), where instead of the random forest algorithm, trees from the Gradient Boosted Decision Tree (GBDT) are used for the categorical data encoding. This work empirically demonstrates that the boosted decision trees are a powerful and convenient way to implement non-linear and categorical feature transformations for heterogeneous data. The DeepGBM framework (Ke et al., 2019) further evolved the idea of distilling knowledge from decision trees leaf index by encoding them using a neural network for online learning tasks.

This leaf embedding approach received much attention. However, the leaf indices' from decision tree embeddings do not fully represent the structure of the decision tree. Hence, each boosted tree is treated as a new *meta categorical feature*. Bruch et al. (2020) proposed a gradient-descent-based strategy that exploits the decision tree structure to propagate gradients in the learning process. Instead, our approach has a strong focus on the exploitation of the tree structures for the data transformation process, that is, the transformation of the heterogeneous tabular data into homogeneous vectors that are especially suited for deep learning but also other machine learning techniques. The observation that local Boolean features can be quite informative for global modeling is also reported in (Pedapati et al., 2020), where the authors exploit sparse local contrastive explanations of a black-box model to obtain custom Boolean features. A globally transparent model is then trained on the Boolean features only; empirically, the global model shows a predictive performance that is only slightly worse than that of state-of-the-art approaches.

In summary, in contrast to state-of-the-art methods that exploit decision tree structures and mainly focus on leaf indices, DeepTLF *utilizes the whole decision tree structure* from a GBDT model, and it furthermore *considers the representation of each feature independently* in the information distillation process. The framework proposed in this work combines the advantages of gradient boosted trees (such as handling different scales, different attribute types, missing values, outliers, and many more) with the learning flexibility of neural networks to achieve excellent predictive performance.

## C  TREEDRIVENENCODER ALGORITHM

In this section, we present the psedo code for our TreeDrivenEncoder algorithm (Alg. 1).

## D  REPRODUCIBILITY DETAILS

We use the following implementation of baseline ML models: kNN, RF, LR, algorithms are from the widely used open-source machine learning python library Scikit-Learn Pedregosa et al. (2011), for GBDT we select the python version of distributed gradient boosting library XGBoost Chen & Guestrin (2016), RLN, we use the official TensorFlow implementation from the GitHub repository[1], we use well-tested PyTorch implementation of TabNet[2]. We use the official implementation of

---

[1]https://github.com/irashavitt/regularization_learning_networks
[2]https://github.com/dreamquark-ai/tabnet

**Algorithm 1** For a GBDT model $\mathcal{T}$ and an instance $\mathbf{x}$ from the underlying dataset, the *TreeDrivenEncoder* procedure visits the inner nodes of each $T \in \mathcal{T}$ (in a breadth-first search manner) and exploits their Boolean functions to construct a binary vector according to the feature values of $\mathbf{x}$.

**procedure** TREEDRIVENENCODER($\mathbf{x}, \mathcal{T}$))
    $\mathbf{x}^b$ vector of length 0
    **for** tree $T \in \mathcal{T}$ **do**
        $\mathbf{u}$ vector of length $|T|$                $\triangleright$ *binary vector we aim to construct*
        $i := 0$                            $\triangleright$ *position index in the binary vector*
        $Q := \emptyset$ an empty queue
        $Q$.enqueue(T.root)
        **while** $Q$.notEmpty **do**
            $v := Q$.dequeue()
            $x :=$ getFeatureValue($\mathbf{x}, v$) $\triangleright$ *get from $\mathbf{x}$ the value of the feature that is evaluated at $v$*
            **if** $v$.evaluate($x$) == true **then**            $\triangleright$ *evaluate $x$ at $v$*
                add $\mathbf{u}(i) = 1$
            **else**
                add $\mathbf{u}(i) = 0$
            **end if**
            i++
            **for all** children $v'$ of $v$ **do**
                **if** $v'$ is an inner node ($v' \in V_I$) **then**
                    $Q$.enqueue($v'$)
                **end if**
            **end for**
        **end while**
        $\mathbf{x}^b =$ concat($\mathbf{x}^b, \mathbf{u}$)
    **end for**
    **return** $\mathbf{x}^b$
**end procedure**

Net-DNF[3]. We use the official PyTorch implementation of NODE from the GitHub repository[4], we adapt the official implementation of DeepGBM[5] to our need using the PyTorch framework. VIME[6] and TabTansformer[7] implementations are from official GitHub repositories. DeepTLF, for the encoding part, we employ the XGBoost[8] implementation of the GBDT algorithm; for the deep learning part, PyTorch (Paszke et al., 2017) is used. Additionally, we will provide a TensorFlow implementation. Note, other implementations of GBDT can be used as well. We select the AdaBelief Optimizer Zhuang et al. (2020) for proposed framework.

For the hyper-parameter selection task for all baseline, we apply the tree-structured parzen estimator (TPE) optimization algorithmBergstra et al. (2011) using the HyperOpt library Bergstra et al. (2013).

**t-SNE Experiment** For the t-SNE experiments, we normalized the original datasets. We did not pre-procces the encoded homogeneous dataset after the TreeDrivenEncoder transformation.

**Computing infrastructure.** Out experimental setup for all experiments has two RTX2080Ti GPUs and a single CPU AMD 3960X 24-Core.

## D.1 DATASETS DESCRIPTION

Among these, the *HIGGS* dataset, which stems from experimental physics, is the largest dataset in our evaluation. As an exemplary dataset from the financial industry, we include the dataset *defaults of clients*, which contains information on default payments, demographic factors, credit data, history

---

[3]https://github.com/amramabutbul/DisjunctiveNormalFormNet
[4]https://github.com/Qwicen/node
[5]https://github.com/motefly/DeepGBM
[6]https://github.com/jsyoon0823/VIME
[7]https://github.com/lucidrains/tab-transformer-pytorch
[8]https://xgboost.readthedocs.io/en/latest/

Table 4: URLs for datasets of the study.

|     | Dataset                    | URL                                                                          |
|-----|----------------------------|------------------------------------------------------------------------------|
| D1  | HIGGS                      | https://archive.ics.uci.edu/ml/datasets/HIGGS                                |
| D2  | Default of clients         | https://www.kaggle.com/uciml/default-of-credit-card-clients-dataset          |
| D3  | Telecom churn              | https://www.kaggle.com/c/zillow-prize-1                                      |
| D4  | Zillow                     | https://www.kaggle.com/neuromusic/avocado-prices                             |
| D5  | Avocado prices             | https://www.kaggle.com/blastchar/telco-customer-churn                        |
| D6  | California housing         | https://www.kaggle.com/camnugent/california-housing-prices                   |
| D7  | E-commerce clothing reviews| https://www.kaggle.com/nicapotato/womens-ecommerce-clothing-reviews          |

of payment, and bill Statements of credit card clients in Taiwan from April 2005 to September 2005. In addition, the *Zillow* dataset represents typical heterogeneous data from the real estate sector. It is important to emphasize that in this dataset around 47 % of the data inputs are missing values. The avocado dataset is another representative of tabular datasets, which provides historical data on avocado prices. The *telecom churn* dataset presents customer data of different feature types with the goal to estimate the behavior of a customer. Lastly, we employ the *California housing* dataset which contains information about house pricing in 1990. *E-commerce clothing reviews* dataset (Agarap, 2018) has multiple data modalities - text and tabular data.

All these datasets are collected from real-world problems and contain numerical as well as categorical data. Moreover, these datasets are freely available online and common in tabular data processing: each dataset was previously featured in multiple published studies. We deliberately chose these six datasets to cover different domain areas (web, natural sciences, etc.), tasks (classification and regression), and different dataset sizes.

We prepossessed the data in the same way for every machine learning model by applying standard normalization. For the linear regression, logistic regression, and models based on neural networks, the missing values were substituted with zeros since these methods cannot handle them otherwise.

