# OpenReview forum: "Robust Deep Neural Networks for Heterogeneous Tabular Data "
_ICLR.cc/2022/Conference — ICLR 2022 Submitted_

### Official Review · Reviewer_gWeP · 2021-11-02

**Correctness:** 3
**Technical Novelty And Significance:** 3
**Empirical Novelty And Significance:** 3
**Recommendation:** 5
**Confidence:** 5

**Main Review:**

strength:
1. the paper is clearly written, and easy to follow.
2. the proposed idea is simple and easy to understand.
3. the experiment part results look promising.

weakness & question:
1. some details in experiments are not clear. For example, the used hyper-parameters, especially for GBDT. And is the results run by one hyper-parameters, and by hyper-parameter search tools? To get a fair comparison, a throughout hyper-parameter search is needed.
2. Can you also provide the AUC metric for the binary classification tasks?
3. Are there any multiclass tasks?
4. Do you think it is possible to get rid of GBDT (as a feature extractor), and independently design a feature extractor, and perform a similar performance?

I will boost my score if the experimental details are provided and reasonable.

**Summary Of The Paper:**

This paper proposes an encoding method (to encode structural information in GBDT) for DNN with tabular data, to handle the heterogeneous features. The method is simple, and the results look very promising.


**Summary Of The Review:**

This paper is clearly written, and the proposed method is simple, while seems to work. But some details of the experiment, especially the hyper-parameters, are missed.

---

> ### Author Response · Authors · 2021-11-19
> **Thank you!**
>
> Dear Reviewer, we thank you for the time you spend reviewing our work. We are sorry for the delay with our reply; we hope you will still find the time to read it. Also, we are glad that you noticed the strength of our work.
>
> ------
> We will address the questions in the order they are posted.
>
> > _some details in experiments are not clear. For example, the used hyper-parameters, especially for GBDT. And is the results run by one hyper-parameters, and by hyper-parameter search tools? To get a fair comparison, a throughout hyper-parameter search is needed._
>
> We are sorry for not including the hyper-parameters ranges for baseline models. In the deadline rush, we overlook that. But, we will have them in the revised version of our work.
> We believe that hyper-parameters tunning, especially for the tabular data format, is extremely important. Therefore, a significant amount of the time was dedicated to this step. We are also going to publish the script, which we used to be fully transparent.
> We used the hyperopt library for that; it is well-known and accepted by the community of researchers and data scientist tools for the hyper-parameters search.
> We also would like to share the ranges for tuning the GBDT baseline here:
> max_depth {2,3,4,5,6,7,8,9},
> learning_rage - (0.1, 0.5)
> Alpha, gamma - (0, 1)
>
> > _Can you also provide the AUC metric for the binary classification tasks?_
>
> Since we are investigating DL methods, the binary cross-entropy (log loss) metric is the most common evaluation criteria from literature after the accuracy for these models. Also, the log loss metric does not suffer from misordering like the AUC, besides in many real-world applications, the calibrated probabilities are essential. Nevertheless, we present a comparison of DNN and DeepTLF models using validation curves in Fig. 7, Fig. 8, Fig. 9, Fig. 10. We kindly ask the Reviewer to look at them.
>
> Another aspect is that DL-based models cannot optimize the ROC AUC metric directly. There are some approximations of the metic, but the majority of the DL-based algorithms for classification tasks use the log loss metic. Therefore, for the DL models comparison, using the binary cross-entropy is more intuitive, in our opinion.
>
> Furthermore, we tuned hyper-parameters of all baseline to optimize the binary cross-entropy metric. Thus, it will be very challenging in the discussion window to re-run all the experiments optimizing the ROCAUC metric.
>
> However, in case of the paper acceptance, we can guarantee that we will provide results using the ROCAUC metric for each used baseline in our work.
>
> > _Are there any multiclass tasks?_
>
> Unfortunately, we have not yet tested it in the multiclass setting. However, we will try to include it in the revised version, and if the paper is accepted will introduce the multiclass benchmark in our work. Thank you for the suggestion.
>
> Also, due to the flexibility of the proposed framework, the DeepTLF model can be easily adapted to multiclass tasks.
>
> > _Do you think it is possible to get rid of GBDT (as a feature extractor), and independently design a feature extractor, and perform a similar performance?_
>
> We believe it is possible; for example, one can utilize the DNN version of a decision tree from NODE or Net-DNF models.  However, in this case, we will lose the power(s) of the GBDT algorithm in preprocessing (handling missing values, noisy data, etc.), and since the GBDT is a sequential algorithm, which allows the better "coverage" of a tabular dataset, thus more robust data distillation. However, we still think it is an exciting question and should be addressed in the follow-up works.
>
> ------
> Dear Reviewer, we thank you again for your suggestions, we will incorporate them into the paper. If you have a follow-up question, we will be happy to answer them.

---

> > ### Comment · Reviewer_gWeP · 2021-11-26
> > **Thank you for the response**
> >
> > Due to the deadline rush, and no revisions are made during reviewing, I think this paper is still not ready for publication. So I will keep the score unchanged. Hopefully, you can provide more details in the next version.

---

### Official Review · Reviewer_vaip · 2021-11-02

**Correctness:** 3
**Technical Novelty And Significance:** 3
**Empirical Novelty And Significance:** 3
**Recommendation:** 6
**Confidence:** 4

**Main Review:**

The idea is nice and quite elegant, leveraging the flexibility of tree based methods for use in deep learning. To my knowledge, this is a novel idea, although it may not be super obscure. At least I did not came across any literature where this approach had been followed.

The paper is well written and easy to follow. If we deduct the recap of gradient boosting and the formal definition of a decision tree, the actual description of the approach is just about one page. Large part of the paper and the appendix is dedicated to extensive experiments, comparing the approach to simple baselines and dedicated tabular data methods.

The major claims made in the paper are supported
- *DeepTLF can preserve most of the information that is contained in the original data.* I think this is shown with the experiments (Table 2) in so far as DeepTLF yield state of the art performance on all datasets
 - *it dramatically speeds up the data preprocessing time (Fig 1, caption)* Well, yes, once the tree is trained no further explicit preprocessing (data scaling and handling missing values) is required as this is moved into the decision functions.

I have three bigger concerns:
1. The comparison with DNN in Table 2 may not be fair because DNN has 3 fully connected layers while DeepTLF has 4 fully connected layers. Depending on the number of decision functions, the first layer in DeepTLF may also have more parameters than DNN. The architecture of DNN is not further described. Are there embedding layers for categorical columns  or are they treated numerically? How are missing values treated? That can make a big difference to the performance of DNN.
2. Table 3 should include all(!) competitors, in particular DNN. For the training time of DeepTLF it should be made explicit how much time is spend on training the tree ensemble (is it the same as GBDT, second line) and how much for training the network. For data pre-processing time it should be made clear what time is required to prepare the dataset for training (e.g. computing averages and variances or inferring candidates for missing values) and how much preprocessing time is required to do inference on a new and unseen sample (e.g. scale fields appropriately, fill in missing values). Both should be set into relation to training time and inference time respectively.
3. I'm wondering what I could do if there are no classification or regression targets to train a tree ensemble in the first place (e.g. for  unsupervised or self-supervised learning approaches) Have the authors done any experiments in this regard? At least this questions should be addressed in the discussion.


#### Minor issues
- Fig 2, caption: $\mathbf x_i \in \mathbb R\times\mathbb R\times\\{red,-\\}\times\mathbb R\$
similar in Definition 1
- p5,l15: *"It is only important that the same strategy is used across all decision trees and vectors.”* why is that? Isn’t it just important that the order of $\tilde\mu_v$ is the same for all $\mathbf x$?
- p5,l28: $\mathbf x=(x_1,…,x_d)^\top$, Euklidean vectors are column vectors by convention, similar in (8)
- how to deal with duplicate $\tilde\mu_v$ ?





**Summary Of The Paper:**

For the problem of learning (supervised classification and regression) on tabular data, the authors propose to used the decision functions of tree based ensemble methods as input features for a deep neural network (DNN). Tabular data is typically heterogenous, that is the data come from different modalities (continuous, categorical, missing values). As decision trees are very well tailored to such data the authors show that with their method (DeepTLF) no further preprocessing or data cleansing is necessary (apart from what's being performed in the decision tree). On numerous experiments on 7 public tabular datasets, they show that their method outperforms different baseline and competitive approaches, thereby supporting the claim that the tree based input features preserve sufficiently rich information of the input data.

**Summary Of The Review:**

Interesting and to my knowledge novel idea to use decision functions of tree ensembles as input features to neural networks. The experiments show that this method works, but are not 100% conclusive if it is superior in all aspects, in particular comparing to DNN (see my concerns above).
I rate the paper *"6: marginally above the acceptance threshold"* and am willing to increase my rating if my concerns are fully addressed.

---

> ### Author Response · Authors · 2021-11-18
> **Thank you!**
>
> Dear Reviewer,
> first of all, thank you for your thorough review and for seeing the potential of our idea. We value the time you spent reviewing our work and your remarks since we know how demanding the schedule can be.
>
> > _The comparison with DNN in Table 2 may not be fair because DNN has 3 fully connected layers while DeepTLF has 4 fully connected layers._
>
> We are sorry for providing incorrect information in the paper. In the deadline rush, we forget to correct the values. To do the fair evaluation, we select the same architectures for the DNN baseline and DeepTLF framework - 4 FC layers with 2 DropOut layers. As we wrote in the captions of Figures 4,5,6,7,8 - the DNN model is identical in training and architecture to the DeepTLF’s DNN part. We again apologize for providing incorrect information; we will correct it.
>
> We would like to add here that we observed more “deeper” architectures perform better on the encoded data from decision trees; we will add this to the discussion.
>
> > _Depending on the number of decision functions, the first layer in DeepTLF may also have more parameters than DNN. The architecture of DNN is not further described._
>
> We are sorry for not providing the exact architecture of the DL part of the DeepTLF and DNN baseline. We will add it to the revised version. Meanwhile, please find the exact parameters of the DL part of the DeepTLF.
>
> | Layer             | # Neurons |
> |-------------------|-----------|
> | Fully-Connected   | 384       |
> | SWISH activation  |           |
> | DropOut           | 0.23      |
> | Fully-Connected   | 64        |
> | SWISH activation  |           |
> | DropOut           |       0.23    |
> | Fully-Connected   | 32        |
> | SWISH activation  |           |
> | Fully-Connected   | 1 or 2   |
>
> > _Are there embedding layers for categorical columns or are they treated numerically?_
>
> We do not use any embedding layers; yes, they are treated numerically.
>
> > _How are missing values treated? That can make a big difference to the performance of DNN._
>
> We didn't preprocess the missing value only for two algorithms: GBDT and DeepTLF. We replace missing values with zeros for all other baselines (LR, RF, TabNet, … ). We also used the z-normalization with all baseline except RF, GBDT, and DeepTLF.
> We agree that data preprocessing largely affects the performance of deep neural networks, especially on tabular data. Therefore, we're very caution with the preprocessing and treated the value consistently across all datasets and baselines.
>
> > _Table 3 should include all(!) competitors_
>
> We apologize for not including all the baseline into Table 3; we will do the experiments for the revision; from our observations, the DeepTLF shows the fastest performance among the specialized DL architectures for tabular data.
>
> Also, we must state that the GBDT (XGBoost) is a well-optimized algorithm and is mainly written in C/C++ language. Other DL frameworks such as TabNet has already multiple version. Therefore the benchmark is not 100% fair. Because the current version of the DeepTLF is not yet well optimized in terms of the train and inference time… However, we do provide a multiprocessing version of the endocrine part already.
>
> We kindly ask the Reviewer to give us some time for the table preparation. We will provide it in 1-2 days.
>
> >_I'm wondering what I could do if there are no classification or regression targets to train a tree ensemble in the first place (e.g. for unsupervised or self-supervised learning approaches) Have the authors done any experiments in this regard? At least this questions should be addressed in the discussion._
>
> We thank you for asking these interesting questions! Indeed we thought about SSL training for the DeepTLF framework. And we have multiple ideas:
>
> A way for that is to use one or several variables from a dataset as target because. Because we can assume that there are latent connections between random variables and the target in a dataset. Also, multiple GBDT models can be trained with different targets, either regression or classification, and then we combine these encoded vectors together.  We also had an idea of trying the isolated forest algorithm, which is mainly used for unsupervised anomaly detection tasks.
>
> We did a small experiment in this regard, and the results are comparable with other baselines (however not good as the supervised way).  We have not included them due to the time limitations and since we have more questions than answers. But we are working in this direction. Furthermore, we believe the encoded data representation is easier to generate using GANs. So, we are investigating this area as well.
>
> Thank you for minor issues, we will correct the manuscript.
>
> --------
> We would like to use the opportunity and thank you again for the time you spent reviewing our article, and we are also very grateful for the feedback quality.
>
> We hope we have answered all your concerns. If you still have questions, we'll be happy to answer them.

---

> > ### Comment · Reviewer_vaip · 2021-12-01
> > **Thanks for your response**
> >
> > The authors have addressed my concerns in their response but for some reason did not submit a revised version of the paper. I, thus, keep my rating as it is: "6: marginally above the acceptance threshold"

---

### Official Review · Reviewer_Q6we · 2021-11-02

**Correctness:** 3
**Technical Novelty And Significance:** 2
**Empirical Novelty And Significance:** 2
**Recommendation:** 3
**Confidence:** 4

**Main Review:**

Strengths:
- The paper proposes a very simple idea (i.e. Use a two-stage feature derivation process for tabular data), which outperforms many contemporary large-scale neural network models for tabular data for several real-world datasets.
- Comparison with contemporary tabular neural nets are appreciated.

Weaknesses:
- The technical novelty is quite limited, since all the authors do is train GBDT first on the entire training set, extracts the internal node values as a feature representation, use those feature representations as new training samples to train a downstream neural network. As discussed in the paper, if the whole process was done end-to-end, it would have been more interesting.
- Experiment details are missing/incomplete. D1, D2, D3 are used for binary classification, and the authors do not discuss the ratio between positive and negative samples (class imbalance). Also, for binary classification, researchers typically use AUROC or AUPRC due to class imbalance, but the authors use cross entropy loss instead.
- The proposed method barely outperforms GBDT in all datasets except for D1. The authors claim that they used a small DNN model for DeepTLF to demonstrate the power of their approach, but since that power is not observed in Table 2, it would have been more convincing if the authors had used several variations of DeepTLF (e.g. deeper DNN, Transformer) and show the power of DeepTLF.
- In Table 3, it is not clear whether the training time measure a single minibatch update, or a single epoch, or the entire training process.
- Is there any reason D7 was not included in Table 2?
- The authors state that D7 consists of textual and tabular data. Does this mean that the cell values of the table can either be a continuous value, categorical value, discrete value, and word tokens? It is not clear how GDBT would be able to process such table. If not, then what does D7 look like as a multimodal tabular data?

**Summary Of The Paper:**

This paper proposes DeepTLF, a new framework for prediction tasks using tabular data. DeepTLF first trains gradient boosted decision trees (GBDT) using the entire tabular training samples. Then it uses node values as the input the neural network predictor (e.g. classifier or regressor) for the actual prediction phase. In essence, the authors rely on GBDT's capability to handle heterogeneous tabular data with potentially many missing values to derive a binary representation of the given sample, and feed that representation to a downstream neural network. DeepTLF was able to demonstrate superior performance in real-world tabular datasets.

**Summary Of The Review:**

The proposed method, DeepTLF, is a very straightforward combination of GDBT and DNN, which seems to outperform all modern tabular neural nets, which I find very interesting. However, the technical novelty is quite limited, and considerable amount of experiment details are missing, therefore making it difficult to accurately evaluate the paper.

---

> ### Author Response · Authors · 2021-11-17
> **Response to the Reviewer Q6we [1]**
>
> **Due to the characters number limitation, we had to split the official comment into two messages. We apologize for the inconveniences.**
>
> Dear Reviewer, thank you for your review. We value the time you spent reviewing our work and your remarks.
>
> > _The technical novelty is quite limited, since all the authors do is train GBDT first on the entire training set, extracts the internal node values as a feature representation..._
>
> We do not train the GBDT on train and test datasets together since we assume the real-world scenario - test labels are not available. Also, as we stated in Subsection 4.2, “Main benchmark” we utilize the (stratified) 5-fold cross-validation scheme, which allows us to evaluate the methods on the whole dataset. We also believe that the training on the entire dataset will create the data leak problem. We apologize for this misunderstanding; we will add extra sentences into the caption of the table and the body of the paper in the revised version.
>
> > _As discussed in the paper, if the whole process was done end-to-end, it would have been more interesting._
>
> We agree the end-to-end deep learning version of the proposed framework will find its niche in the community. However, by replacing the GBDT algorithm, we will lose the preprocising robustness of the DeepTLF framework. To be more exact, the GBDT algorithm offers the following advantages -> internal handling of categorical variables, missing values, noisy values. As shown by multiple works [1,2], such tabular data representations broadly impact the performance of the deep neural network. And we also demonstrate this issue in numerous experiments, such as inconsistent data having a large impact on the performance of deep neural networks (we kindly ask the Reviewer to look at Fig. 4, Fig. 9).
>
> > _Experiment details are missing/incomplete. D1, D2, D3 are used for binary classification, and the authors do not discuss the ratio between positive and negative samples (class imbalance)._
>
> We apologize for not providing the positive and negative class ratios, we add them to the revised version.
>
> |    | Dataset                     | Negative class (0) % | Positive class (1) % |
> |----|-----------------------------|----------------------|----------------------|
> | D1 | HIGGS                       | 0.47                 | 0.53                 |
> | D2 | Default of Clients          | 0.778                | 0.221                |
> | D3 | Telecom churn               | 0.712                | 0.288                |
> | D7 | E-commerce clothing reviews | 0.178                | 0.822                |
>
> > _Also, for binary classification, researchers typically use AUROC or AUPRC due to class imbalance, but the authors use cross entropy loss instead._
>
> We appreciate your suggestion about the AUROC and AUPRC metrics, and we will try to include the benchmarks using these metrics in the revised version, however, due to large numbers of experiments and current load with other projects, we cannot guarantee that. If the paper is accepted we will definitely include benchmark results using the aforementioned metrics.
>
>
> > _The proposed method barely outperforms GBDT in all datasets except for D1. The authors claim that they used a small DNN model for DeepTLF to demonstrate the power of their approach, but since that power is not observed in Table 2_
>
> We agree, however, our evaluation strategy is challenging since we use the (stratified) 5-fold cross-validation strategy. Also, in comparison to a DNN, our approach, on average across all datasets, shows 19.6 % improvement. We provide validation curves in the Appendix,
> Also, the DeepTLF shows solid performance under corrupted data experiments: missing values, noisy data, and noisy labels. We believe this characteristic is highly desirable for critical real-world applications.
>
> >_it would have been more convincing if the authors had used several variations of DeepTLF (e.g. deeper DNN, Transformer) and show the power of DeepTLF._
>
> We want to demonstrate that the DeepTLF framework performance well even using a relatively simple architecture and not due to the power of the Transformers.
>
> >_In Table 3, it is not clear whether the training time measure a single minibatch update, or a single epoch, or the entire training process._
>
> We used the whole available dataset for training and inference steps. The information about the number of the samples is available in the caption of Table 3.
>
> > _Is there any reason D7 was not included in Table 2?_
>
> Yes, the D7 is the multimodal dataset, it has two modalities: tabular and text data. Not all baseline are able to work with this setting (without an extensive data preprocessing step).

---

> > ### Author Response · Authors · 2021-11-17
> > **Response to the Reviewer Q6we [2]**
> >
> > **The second part of our official comment. We apologize for the inconveniences.**
> >
> > >_The authors state that D7 consists of textual and tabular data. Does this mean that the cell values of the table can either be a continuous value, categorical value, discrete value, and word tokens? It is not clear how GDBT would be able to process such table. If not, then what does D7 look like as a multimodal tabular data?_
> >
> > In the following, we provide a first row from the E-commerce clothing reviews dataset:
> >
> > |    |   Clothing ID |   Age |   Title | Review Text                                           |   Rating |   Recommended IND |   Positive Feedback Count | Division Name   | Department Name   | Class Name   |
> > |---:|--------------:|------:|--------:|:------------------------------------------------------|---------:|------------------:|--------------------------:|:----------------|:------------------|:-------------|
> > |  |           767 |    33 |     nan | Absolutely wonderful - silky and sexy and comfortable |        4 |                 1 |                         0 | Initmates       | Intimate          | Intimates    |
> >
> > It can be seen that the data have continuous, discrete, and text variables. Also, the D7 dataset contains the missing values.
> >
> > ---------
> >
> > Dear Reviewer, we believe our approach demonstrates solid performance in multiple challenging experiments (in comparison to state-of-the-art models and under corrupted data). And we show that DeepTLF can be applied in real-world scenarios due to its robustness to the data inconsistencies.  If you think our response addresses your concerns about our work, please consider revising your score.
> >
> > [1] Hancock, John T., and Taghi M. Khoshgoftaar. "Survey on categorical data for neural networks." Journal of Big Data 7.1 (2020): 1-41.
> >
> > [2] Sánchez-Morales, Adrián, et al. "Improving deep learning performance with missing values via deletion and compensation." Neural Computing and Applications 32.17 (2020): 13233-13244.

---

> > > ### Comment · Reviewer_Q6we · 2021-12-01
> > > **Thank you for the response**
> > >
> > > I appreciate the authors posting the response, but most of my main concerns remain, such as lack of novelty (I stated that the authors are training GBDT on the entire "training set", not the entire "dataset", but the author response suggests that they either misread my review, or intentionally misinterpreted my statement) and a strange choice of evaluation metric (not using AUROC or AUPRC, even though they can be easily calculated with a trained model). Furthermore, considering the authors have failed to provide the revised manuscript, I keep my score.

---

### Author Response · Authors · 2021-11-28
**General Response**

### Dear Reviewers,


We thank all our Reviewers for their time, detailed suggestions, and constructive comments.

In this general response we would like to address the common concerns about our work.

------

The main contribution of our work is the demonstration that deep neural networks are capable to perform robustly and accurately on tabular data without the need for complex DL architectures. We confirm that using multiple challenging experiments on various data set even with different modalities.  This brings us to the conclusion that the learning representations are one of the key charatestics for robust deep tabular learning. We propose to _transform heterogeneous tabular data into homogeneous tabular data_ using boosted decision trees.

We also believe that the gradient boosted decision trees algorithm is a reliable method for learning the homogenous representation of tabular data. Since, it brings the robustness, e.g., handling inconsistent data samples with missing and noisy values, plus it help to provide robust representation even on the small data by performing local feature engineering (we extract relationship of the data sample to a certain threshold).  Therefore, the fully-differentiable version of the DeepTLF approach might lose this valuable characteristics.

We provide multiple experiment with the most commonly used metrics in research or industry: Accuracy, Binary Cross-Entropy, ROC AUC, and MSE. We kindly asking Reviewer and Reader to see the following figures: Fig. 4, 8, 9, 10, 11.
For the main benchmark on classification datasets (Tab. 2) we utilise the binary cross-entropy since the most of the all DL-based algorithms are optimising it, therefore the comparison using this metric is a fair evaluation. Besides, the binary cross-entropy allows to see the confidence of the classifier, where ROC AUC mostly shows the ranking.

---

We thank Reviewers again for their time and thankful comments, we would like to also state that all your comments will be addressed in the next version. If Reviewers have any other questions or concerns, please let us know and we will do our best to resolve them.

---

### Decision · Program_Chairs · 2022-01-20

**Decision:**

Reject

**Comment:**

The paper introduces a method called DeepTLF that handles heterogeneous tabular data by using GBDT as an encoder for a DNN.

The paper is clearly written and the method works as intended.

There is however the issue of novelty (raised by Q6we). The method indeed relies of the capacity of GBDT to represent the data, the internal node values are used as features to train a downstream neural network. This process is straightforward, which is good from an application perspective, though the paper offers limited insights to the community from a scientific perspective.

Another reviewer concern was that of incompleteness of experiments and lack of certain details (reviewers vaip and gWeP). This was answered in the rebuttal, which the reviewers acknowledged, however, the authors did not provide a revised version of the manuscript, when ICLR in fact allowed (and actually encouraged) revised versions to be submitted by Nov 22. Without a revised version, it is difficult for the reviewers to assess whether the text in the final manuscript will actually accurately reflect the changes they suggested. This justifiably caused two of the reviewers to keep their original scores (they explicitly stated the lack of an updated manuscript as the reason).

Given lack of an update, coupled with the issue of novelty, I conclude the paper is not ready to be accepted in its current form.